# Efficacy of Arabica Versus Robusta Coffee in Improving Weight, Insulin Resistance, and Liver Steatosis in a Rat Model of Type-2 Diabetes

**DOI:** 10.3390/nu11092074

**Published:** 2019-09-03

**Authors:** Pedram Shokouh, Per B Jeppesen, Christine B Christiansen, Fredrik B Mellbye, Kjeld Hermansen, Søren Gregersen

**Affiliations:** 1Department of Endocrinology and Internal Medicine, Aarhus University Hospital, Palle Juul-Jensens Boulevard 99, 8200 Aarhus N, Denmark; 2The Danish Diabetes Academy, Odense University Hospital, Kløvervænget 10, 5000 Odense C, Denmark; 3Department of Clinical Medicine, Aarhus University, Palle Juul-Jensens Boulevard 165, 8200 Aarhus N, Denmark; 4Steno Diabetes Center, Palle Juul-Jensens Boulevard 165, 8200 Aarhus N, Denmark

**Keywords:** coffee, insulin resistance, type 2 diabetes mellitus, liver steatosis, gene expression

## Abstract

The effects of chronic coffee exposure in models of type 2 diabetes mellitus (T2D) is scarcely studied, and the efficacy of the main coffee species has never been compared. We tested the hypothesis that long-term consumption of arabica and robusta coffee may differentially delay and affect T2D development in Zucker diabetic fatty rats. Three study groups received either chow mixed with arabica or robusta instant coffee (1.8% *w*/*w*) or unsupplemented chow food for 10 weeks. Both coffee species reduced liver triglyceride content and area under the curve of fasting and postprandial insulin. At study end, plasma adiponectin, total cholesterol and high density lipoprotein levels were higher in the robust group compared with both arabica and control groups. The liver gene expression of Glucose-6-phosphatase, catalytic subunit (G6pc) and Mechanistic target of rapamycin (mTOR) in robusta and Cpt1a in both coffee groups was downregulated. In conclusion, long-term consumption of both coffee species reduced weight gain and liver steatosis and improved insulin sensitivity in a rat model of T2D. Robusta coffee was seemingly superior to arabica coffee with respect to effects on lipid profile, adiponectin level and hepatic gene expression.

## 1. Introduction

Decades of research have unveiled some favorable effects of coffee in a wide array of health conditions [1] and several studies indicate beneficial effects of coffee towards development of type 2 diabetes mellitus (T2D) [2].

The coffee plant belongs to the *Coffea* L. genus of the family Rubiaceae with over 120 species. *Coffea arabica* L. and *Coffea canephora* L. (commonly referred to as robusta) are the main species with commercial importance. The chemical composition of coffee brews depends on the coffee species. These differences contribute to a characteristic flavour and quality of coffee beverage made from each species. Typically, green robusta seeds contain almost twice as much caffeine, more chlorogenic acids, and less trigonelline than arabica per weight unit. However, the chemical composition varies greatly depending on the quality of the coffee and roasting degree [3,4,5,6,7]. The question arises if this translates into differential health effects of administration of coffee made from robusta coffee compared to arabica coffee. Differential effects of coffee seed oil from arabica and robusta in humans seem to exist, e.g., on serum lipids and liver enzymes [8,9]; however, in vivo effects of whole coffee bean extracts from the two species have never been compared. Learning more about any potential differences may increase our understanding of the distinct metabolic effects of bioactive compounds in coffee.

Several short term studies have indicated beneficial effects of coffee or coffee components in regards to T2D. Thirty days of administration of coffee arabica to Zucker fa/fa improved triglycerides, low-density lipoprotein (LDL) cholesterol, and glucose levels compared to Zucker fa/fa control rats, with the levels corresponding to the levels in Zucker +/+ rats. However, glucose levels improved only at intervention week 3 and a reduction in low-density lipoprotein (HDL) cholesterol was found [10]. In addition, five weeks intervention with coffee administered to KK-ay mice lowered blood glucose and liver lipids compared to water [11]. Furthermore, lepr db/db mice treated for two weeks with chlorogenic acid showed glucose, insulin, and lipid levels similar to healthy rats and metformin-treated lepr db/db mice. Chlorogenic acid also improved glucose uptake in skeletal muscle and lower liver lipid content was found [12]. Similarly, chlorogenic acid improved levels of blood and liver lipids in comparison to control solution in a 3-weeks intervention study using Zucker fa/fa rats [13]. Furthermore, caffeine [11] as well as trigonelline and nicotinic acid [14] have beneficial effects toward T2D shown by lower blood glucose and liver lipids levels in KK-ay mice.

Based on the previous research on animal models of T2D, we hypothesize that long-term coffee consumption per se has beneficial effects in preventing and/or controlling the metabolic disturbances of T2D and that the efficacy profile of the two coffee species differs. To test these two hypotheses, we used a inbred rat model of T2D, the Zucker diabetic fatty (ZDF) rat, and compared different indices of insulin resistance (IR), plasma lipids, body weight, liver steatosis, and gene expression profile in rats receiving single-species instant coffee during and after the development of T2D features.

## 2. Materials and Methods

### 2.1. Experimental Animals

ZDF rat is an inbred model with a homozygous missense mutation (fa/fa) in the leptin receptor gene (Lepr) resulting in hyperglycaemia after approximately 7 weeks of age [15] and have phenotypic similarities with human T2D. Thirty-three 6-week-old male ZDF rats were purchased from SCANBUR A/S (Karlslunde, Denmark). Rats were caged individually and kept under reversed 12-hour light-dark cycles with free access to regular rat chow (Altromin 1324, Altromin GmbH, Lage, Germany) and tap water. After 1 week of acclimatization, the rats were randomly assigned to control and intervention groups. In one of the ZDF rats in the control group, characteristic hyperglycaemia and insulin deficiency did not develop and, therefore, that rat was excluded from the final analyses. To control for bodyweight differences between intervention groups we employed a pair feeding regimen at the beginning of week 5 after which the intervention groups hereafter received equal amounts of food by weight to the control group based on weekly food consumption measurements. The animals were housed and handled in keeping with the Legislative Decree no. 253, 8 March 2013, under regular inspections and approval of the Danish Animal Experiments Inspectorate (Approved 2015, No. 2012-15-2934-00200).

### 2.2. Study Design and Procedures

Rats were divided into three groups by random labelling (control, arabica, and robusta groups, *n* = 11 per group). Intervention groups received a custom-made Altromin 1324 containing 1.8% *w*/*w* of either arabica or robusta instant coffee for 10 weeks. Instant coffee is based on filtered coffee, thus, it was used as an alternative to filter coffee which is the most commonly used coffee brewing method in Scandinavia [16]. The custom-made single-species Brazilian instant coffee and caffeine measurements were provided by BKI Foods A/S (Højbjerg, Denmark). Robusta seeds originated from Espirito Santo state and the caffeine content was 3.85% of dry weight. Arabica seeds originated from Minas Gerais state in Brazil and the caffeine content was 2.35% of dry weight. With an expected average food intake of 33 g/day [17], the daily instant coffee intake of each rat was 0.597 g. If a ZDF rat’s average weight of 360 g and body surface area (BSA) of 0.05 m^2^ is assumed, 0.597 g/day would be equal to 20.2 g/day of instant coffee per day in an adult human (BSA = 1.69 m^2^).

The rats were weighed before the start of the intervention and every week afterward. Food consumption was monitored weekly by dispensing a fixed amount of food and weighing the leftovers. Blood sampling was carried out by cutting the tip of the tail every 3rd week. One week before the study termination, a standard oral glucose tolerance test (OGTT) was performed by administering 2 g/kg of d-glucose by gastric gavage and collecting blood samples before and 30, 60, 120, 180, and 240 min after the gavage. Rats were fasted for 6 to 8 h before each blood sampling procedure, including the OGTT.

At the end of the study, rats were terminated by cervical dislocation under deep anesthesia induced by intraperitoneal injection of 50 mg/kg of pentobarbital. After blood sampling from the retro-orbital plexus, the liver was harvested and fixed instantaneously by submerging in liquid nitrogen and stored at −80 °C. 

### 2.3. Experimental Measurements

Blood samples were collected in heparin/aprotinin-containing tubes and centrifuged at 4000 RPM at 4 °C for 15 min. Plasma was kept at −80 °C until analysis. The glycaemic level was evaluated by measuring fasting plasma glucose (FPG) every 3rd week using Cobas c111 analyzer (Roche Diagnostics, Mannheim, Germany) and glycated haemoglobin (HbA1C) quantified at baseline and endpoint in whole blood collected in EDTA tubes using the Cobas c111 analyzer. IR was estimated by fasting plasma insulin (quantified by sensitive rat RIA kit (EMD Millipore, Billerica, MA, USA), postprandial plasma glucose and insulin, homeostatic model assessment-insulin resistance (HOMA-IR) calculated using the formula [18]: glucose mmol/L × insulin mIU/L/22.5, and Matsuda insulin sensitivity index using the formula [19]: 10,000/√ ((fasting glucose mg/dL × fasting insulin mIU/L) × (OGTT0-120 mean glucose mg/dL × OGTT0-120 mean insulin mIU/L)). Plasma lipoproteins were measured by the Cobas c111 analyzer before and after the intervention, and free fatty acids (FFAs) by NEFA-HR2 assay (Wako Chemicals GmbH, Neuss, Germany) only at the endpoint. Plasma adiponectin was quantified by AssayMax ELISA kit (Assaypro, St. Charles, MO, USA). For liver triglycerides content measurement, approximately 50 mg of frozen liver tissue was weighed on dry ice and its saponified extract was obtained using a published protocol [20]. Triglyceride concentration was then quantified by the Cobas c111 analyzer.

### 2.4. Quantitative Real-Time Polymerase Chain Reaction

Total RNA of 50 mg of liver tissue prepared as described above was extracted using the RNeasy Minikit (Qiagen GmbH, Hilden, Germany) according to the manufacturer’s instructions. RNA concentration in samples was evaluated by measuring UV absorbance at 260–280 nm (NanoDrop ND-8000 UV−vis spectrophotometer, NanoDrop Technologies, Wilmington, DE, USA).

Quantitative real-time polymerase chain reaction (PCR) on a Fluidigm BioMark System (AROS, Applied Biotechnology A/S, Aarhus, Denmark) was used to compare the expression level of 21 target- and two internal-reference (Hprt1 and Gapdh) genes (Table 1). Custom rat-specific TaqMan assays with IDs listed in Table 1 were used in the analyses. Samples were analyzed using Fluidigm 96.96 Dynamic (Fluidigm catalog no. BMK-M-96.96) arrays with assay triplicates in accord with the manufacturer’s protocol. A total of 100 ng of RNA was used as input in 20-μL reverse transcription reaction using the High Capacity cDNA Reverse Transcription Kit (ABI, PN4368813). The obtained cDNA was amplified using a target-specific assay (diluted 1:100) and TaqMan PreAmp Master mix (2X) (ABI, PN 4391128) in a 14-cycle thermal cycler reaction: 95 °C 10 min and 14 cycles of 95 °C 15 s and 60 °C 4 min. Amplification was performed using the standard conditions: 50 °C 2 min, 95 °C 10 min, and 40 cycles of 95 °C, 15 s, and 60 °C, 1 min. Relative quantification method was used in the analysis of the PCR data reported as the average of the threshold cycles (Ct) triplicates using the ΔCt method with the formula: 2 ^Ct reference gene − Ct target gene^). Based on the reasons explained in the results and discussion sections, we decided to use one of the reference genes instead of their geometric means.

### 2.5. Statistical Analyses

Data are presented as means plus 95% confidence interval (CI) or standard error of the mean (SEM). Study groups were compared by one-way analysis of variance (ANOVA) followed by group-wise post hoc comparisons. The partial eta squared(ηp2) was calculated to estimate the effect size of significant findings. Protected Fisher least significant difference (LSD) test was used as a valid method for three-group comparisons [21]. Where Levene test rejected the equality of variances, an F-test from the Brown–Forsythe statistic was used instead of the one from one-way ANOVA, and Games-Howell post hoc test was employed as an alternative to the LSD test. All the analyses were performed at a two-sided significance level of 0.05 using IBM SPSS Statistics for Windows (Version 22.0, IBM Corporation, Armonk, NY, USA). Area under the curve (AUC) was calculated using GraphPad Prism (version 5.01, GraphPad Software, La Jolla, CA, USA). The rate of changes of selected variables was calculated using the slope function of Microsoft Excel 2013 (Microsoft Corp., Redmond, WA, USA).

## 3. Results

### 3.1. Food Consumption and Body Weight

As illustrated in Figure 1a, both coffee-receiving groups had significantly higher food consumption compared to control from the first measurement. The arabica group had a higher incremental slope of food intake from week 2 to week 4 compared to robusta (*P* = 0.048). At week 4, rats in the arabica group were eating 14.7 g/day (95%CI = 12.0–17.9 g/day) and in the robusta group 7.5 g/day (95%CI = 3.2–11.7 g/day) more food than the control rats (Figure 1a). This phenomenon of inducing an upsurge in calorie intake by coffee has previously been reported in mice fed a high-fat diet [22]. Faced with this situation, we decided to harmonize the food intake of all groups by pairing the coffee groups’ food intake to that of the control rats. This approach could prevent the metabolic effects of coffee from being outweighed by the consequences of overeating. However, with a significant difference between the food consumption and, therefore, the coffee dosage, it would have been invalid to make any comparisons between the intervention groups. After week 4, the rate of weight gain in both arabica and robusta groups declined significantly (*P* = 0.003 and *P* ˂ 0.001, respectively, compared to the control group). Incremental AUC of body weight from week 4 was lower than control in both arabica (−286.0 g/6 weeks, 95%CI = −412.5 to −160.6 g/6 weeks) and robusta (−207.9 g/6 weeks, 95%CI = −333.2 to −82.6 g/6 weeks) groups (F-test *P* < 0.001, *p*η2 = 0.660) without a significant difference between the two intervention groups (Figure 1b).

### 3.2. Glucose Homeostasis

At eight weeks of age the rats already had elevated levels of FPG before being allocated to the interventions. The FPG escalated in all groups over time (Figure 2a). Despite a significant between-group difference at intervention week 3 (F-test *P* = 0.033), the post hoc analysis did not detect any considerable differences at each time point, as well as in AUC or change rates between groups. HbA1C, a measure of average glycaemia, was assessed before and after the intervention. Coffee groups were not different from the control group despite the inequality of variances at the endpoint. Rats receiving robusta coffee, however, had a lower increase in HbA1C levels throughout the study in comparison with the arabica group (Table 2).

### 3.3. Surrogate Indices of Insulin Resistance

We used a surrogate index, HOMA-IR, derived from steady-state conditions, and two dynamic surrogates, OGTT and Matsuda index, to estimate the level of IR at different time points. Fasting insulin was lower in both coffee groups at week 3 (Figure 2b). The initial increase in fasting insulin in control rats was blunted in the coffee groups leading to a lower AUC in both arabica (*P* = 0.004) and robusta (*P* = 0.049) groups (F-test *P* < 0.014, *p*η2 = 0.310). The decline rate of fasting insulin between weeks 3 and 10 was lower than control in both coffee groups (*P* = 0.001 for arabica and *P* = 0.013 for robusta groups). However, the effect of arabica and robusta coffee on fasting insulin was not significantly different. We also did not observe any significant differences in HOMA-IR as illustrated in Figure 2c.

Except for a small but significant difference at time zero, rats in all three groups expressed comparable plasma glucose levels at all time points during the OGTT with no different variance levels (Figure 3a). No differences were noted in the incremental AUCs of the glucose response curves. Both coffee groups expressed a moderated insulin response after the glucose challenge with significantly lower values after 120 min as illustrated in Figure 3b. The AUC of the insulin response in both arabica (*P* = 0.048) and robusta (*P* = 0.022) groups was lower compared to the control group (F-test *P* = 0.047, *p*η2 = 0.253). The Matsuda index, proposed by Matsuda et al. [23], is designed to estimate muscle and liver insulin sensitivity from glucose and insulin data obtained during an OGTT. All three groups in our study had a low Matsuda index at the endpoint, which denotes staggeringly low insulin sensitivity (Figure 3c). Between-group analysis of variance did not detect significant differences in mean Matsuda index of control versus coffee groups.

### 3.4. Plasma Lipids

The main classes of plasma lipoproteins were quantified before and after the intervention period. As expected, the plasma level of all lipoproteins increased compared to baseline. As summarized in Table 2, ANOVA showed higher total cholesterol in robusta group at endpoint compared to both arabica and control groups. HDLs was higher in the robusta group compared to the other two groups (*p*η2 = 0.331). It can be inferred from the comparable triglyceride and LDL levels that HDL was the main contributor to the higher total cholesterol values in the robusta group at the endpoint. Total plasma FFAs levels at endpoint were not statistically different between groups (Table 2). 

### 3.5. Plasma Adipokines

The tendency of higher plasma adiponectin levels in both coffee groups reached a significant level only in the robusta group compared to control (*p*η2 = 0.271) (Table 2). 

### 3.6. Liver Triglyceride Content

In order to have a direct estimate of the degree of liver steatosis (a presentation of nonalcoholic fatty liver disease), liver tissue triglyceride concentration was measured. As it can be seen from Table 2, both coffee species caused a comparable decrease in liver triglyceride content (*p*η2 = 0.374). 

### 3.7. Gene Expression Analyses

Table 1 lists the genes selected for real-time PCR analysis, and Figure 4 depicts the fold-difference in gene expressions of coffee groups relative to the control group. The mRNA of two reference genes, Hprt1 and Gapdh, was amplified concomitantly with the target genes to be used as an internal control; but due to a significant between-group difference in the expression of Gapdh (ANOVA *P* = 0.044), we solely used Hprt1 when normalizing the PCR data. Analysis of 2ΔCt values revealed a significant relative downregulation of Glucose-6-phosphatase, catalytic subunit (G6pc), Cpt1a, and Mechanistic target of rapamycin (mTOR) genes in the robusta group and only Cpt1a gene in the arabica group compared to controls. G6pc mRNA expression in the robusta group was suppressed 2.7 times (SEM = 0.54) compared to the control group (*P* = 0.031). Cpt1a mRNA level was considerably lower in both intervention groups in our study [mean 2ΔCt (SEM) for control = 0.708 (0.061), arabica = 1.059 (0.106), and robusta = 2.519 (0.592), for both coffee groups compared with control *P* = 0.033]. The expression of mTOR was also downregulated in the robusta group compared to in the control group.

## 4. Discussion

In the current study, we investigated the effects of two single-species coffee extracts in ZDF rats, which closely mimic the course and metabolic abnormalities of human T2D. According to the drastic course of changes in beta-cell function and peripheral IR in spontaneous diabetic models, study duration is a crucial factor in any research involving these models. Contrary to the short intervention time (two to five weeks) of comparable studies [10,11,12,13,14], we followed the rats until they reached a state of insulin deficiency. This element might have minimized the between-group differences in metabolic parameters; nonetheless, it presents a more comprehensive picture. The average daily coffee intake in this study was a human-equivalent dosage of approximately 10 cups per day based on the BSA normalization method. This method is generally preferred over a simple conversion based on weight in dose translation between species [24]. In human observational studies, 10 cups of coffee per day showed the maximum protection against T2D [25].

After week 4 when the pair feeding was initiated, the bodyweight remained lower in both the arabica and robusta groups compared to in the control group. The increased food consumption in the coffee treated groups during the first four weeks was very unexpected. It might be characteristic for the animal model used in this study, as the ZDF rats have a defect in the gene encoding the leptin receptor that causes hyperphagia [26]. The combination of coffee and leptin deficiency may prolong their waking state, resulting in increased food intake. Food restriction made the differences in energy expenditure evident, resulting in a significant reduction in body weight. Calorimetric measurements are needed to verify this hypothesis. Weight-modifying effects of coffee, as found in the current study, have also been reported in some previous studies [22,27]. This specific effect is hypothesized to be mediated mainly by the phenolic acid and alkaloid components in coffee. These compounds have been shown to reduce lipogenesis by downregulating the sterol regulatory element-binding protein 1C and accelerating fatty acid uptake and beta-oxidation by activating the adenosine monophosphate kinase pathway to point to a few possible mechanisms [28]. A lower dose of coffee in short-term studies (≤5 weeks) did not alter body weight in diabetic mice [10] and rats [11]. This discrepancy emphasizes the role of treatment duration and dosage in yielding significant effects on body weight. This may be a reflection of the observational findings that the risk of developing T2D in humans decreases 6% per cup of coffee per day. The inverse correlation between T2D and cups of coffee per day is linear up to 10 cups per day which indicates that the effect could be dose-dependent [2].

We were only able to detect a difference in FPG at intervention week 3. This is in contrast with results from short-term studies in KK-Ay mice [11] and Zucker fatty rats [10]. Although different settings and intervention periods make the comparison difficult, it should be considered that ZDF rats develop a more severe form of T2D than Zucker fatty rats in regards to beta-cell deficiency and insulin resistance [29]. The increase in insulin concentration in the control group within the first three weeks might be compensatory for insulin resistance. This increase is not apparent in the treatment groups, indicating a lower degree of insulin resistance compared to the control group. The insulin concentration decreases substantially in all groups in the following weeks, suggesting deficient insulin production. Thus, coffee might protect from insulin resistance to a certain degree, but it is not capable of preventing severe beta-cell failure. Another possibility is that the effect of coffee might decline over time, as Abrahao et al. did not observe any difference in the level of glycaemia between the coffee and the control group at the end of the study, after four intervention weeks [10].

In addition to beta-cell dysfunction, ZDF rats are known to express a progressive degree of IR from an early stage of diabetes. Hyperinsulinemic-euglycaemic clamp technique revealed significantly decreased glucose infusion and peripheral disposal rates in 10-week-old ZDF rats compared to their lean littermates [30]. No glucose clamp data are available to show the trend of changes in later stages of diabetes in ZDF rats. To assess IR we calculated HOMA-IR from FPG and fasting insulin as surrogate markers. The fasting insulin concentrations were lower in the coffee groups compared to the control group at week 3. Having FPG levels similar to the controls with lower insulin secretion indicated a higher insulin sensitivity in both coffee groups. However, the tendency of having lower HOMA-IR levels in the intervention groups at weeks 3 and 6 (Figure 2c) did not reach a significant level, probably due to comparable FPG levels in all groups. In the present study, HOMA-IR should be interpreted with caution. It is suggested that this index of IR cannot be utilized when the insulin-glucose dynamics are disrupted, e.g., in insulin-deficient subjects [31]. We also calculated the incremental AUC for glucose during an OGTT, but observed no differences. These results mirror what Cowan et al. reported from a long-term study on high-fat-fed obese rats [32], but contradict the findings of a similar study in high-fat-fed mice [33]. However, insulin response test conducted along with the OGTT provides more details on the IR status of our rats. In accordance with the fasting measurements, both coffee species allowed rats the ability to maintain comparable levels of blood glucose with lower amounts of insulin. This finding indicates an improved insulin sensitivity in the coffee groups. Insulin-sensitizing effects of coffee have also been reported previously in KK-Ay mice using an insulin tolerance test [11]. A lengthy discussion on which coffee compounds may affect insulin sensitivity is beyond the scope of this discussion; however, it can be briefly mentioned that chlorogenic acid [13], caffeine [11], and trigonelline [14] improved IR in similar study settings. Although coffee improved the insulin concentration following an OGTT, we did not observe any differences in Matsuda indices. We assume that the modest effect of coffee on insulin level was not enough to alter a composite index such as the Matsuda in 17-week-old ZDF rats with advanced beta-cell dysfunction and glucose intolerance.

We demonstrated a positive effect of robusta coffee on total cholesterol and HDL cholesterol compared to control. Knowing that diterpenes are present in negligible quantities in instant coffee [34], and rodents are relatively resistant to their LDL-raising effects [35], no LDL elevation in the coffee groups was expected. Effects of coffee on plasma lipoproteins can be attributed to chlorogenic acids [36] and trigonelline [37] that have shown similar effects before. Robusta coffee also had beneficial effects on plasma adiponectin. Coffee’s adiponectin-increasing efficacy was formerly observed in humans with diabetes [38]. Caffeine is proposed to play a pivotal role, as caffeinated coffee was more potent than decaffeinated coffee in increasing adiponectin levels [39]. Because adiponectin possesses insulin-enhancing and sensitizing properties and positively correlates with plasma HDL levels [40], more potent effects of robusta coffee on HDL and different effects on long-term glycaemic control might have been mediated by higher adiponectin concentrations. Leptin was not measured because it is not reliable in leptin-receptor-deficient models. It correlates with insulin levels rather than with body weight in ZDF rats [41]. Both coffee groups had lower liver triglyceride content. This shows a protective effect of coffee irrespective of the species against liver steatosis in insulin-resistant animals. This finding confirms a previously reported mitigating effect of coffee and caffeine on liver steatosis in KK-Ay mice [11]. Several observational studies in patients with nonalcoholic fatty liver disease also pointed to an inverse association between coffee consumption and steatosis and fibrosis [42].

In our study, we examined the expression of selected genes in liver tissue. G6pc encodes a catalytic subunit of glucose-6-phosphatase, a rate-limiting enzyme, that catalyzes the terminal step in both gluconeogenesis and glycogenolysis. Coffee robusta suppressed G6pc mRNA expression by 2.7 times compared to the controls. This might be due to a direct inhibitory effect or a result of improved hepatic insulin sensitivity and partly explain the superior metabolic efficacy of robusta coffee compared to arabica. It is known that hepatic glucose-6-phosphatase activity is increased in T2D [43], and treatment with either insulin or metformin can downregulate its expression [44]. Cpt1a mRNA codes for carnitine palmitoyltransferase I, a key enzyme in the carnitine-dependent transport of acyl group of long-chain fatty acids through the mitochondrial membrane. Both treatment groups had lower Cpt1a expression. Concomitantly, Cpt1a inhibitors such as Elomoxir and Teglicar have been used as metabolic modulators to treat T2D [45]. These compounds reduced in vivo gluconeogenesis and improved insulin sensitivity [46,47].

The expression of mTOR was reduced in the robusta group compared to in the control group. mTOR is an atypical serine/threonine protein kinase and a part of two signaling complexes mTORC1 and mTORC2. mTOR pathway activity is low during fasting and increased in the liver and muscle of obese rats. Overactivation of mTORC1 promotes inhibitory phosphorylation of insulin receptor substrate-1 that leads to hepatic IR [48]. Modification of Cpt1a and mTOR gene expression by coffee in a type-2 diabetic model that is being reported for the first time here can be an important mechanism by which coffee improves carbohydrate metabolism and reduces the risk of T2D. Although the responsible compounds are not yet known, caffeine would be the first suspect because it inhibits mTORC1 in vitro [49]. Because we did not quantify the protein level of the target genes in this study, our gene expression results await further confirmation.

The current study benefited from using a relevant model, dosage, and administration method. It is worth noting that instant coffee lacks diterpenes, such as cafestol, which have been shown to increase insulin sensitivity and glucose uptake [50]; therefore, it is not unsubstantiated to assume that more pronounced effects can be expected from unfiltered coffee. Discrepancies clearly exist between the human model of T2D and leptin-deficient models. Hyperphagia and IR caused by leptin receptor abnormalities, defective non-shivering thermogenesis, increased activity of lipoprotein lipase, and sex-specific diabetes phenotype are some of these mismatches in ZDF rats that have been reviewed comprehensively by Wang et al. [51].

Our findings provide evidence that both main coffee species, arabica and robusta, can counteract weight gain, reduce hyperinsulinemia, and mitigate liver steatosis in a rodent model of T2D. Our findings could not support the hypothesis that coffee may prevent or delay the development of T2D, however, they indicated its efficacy in modifying related metabolic disturbances. Robusta coffee expressed superior effectiveness in improving plasma lipid profile, long-term glycaemia, and plasma adiponectin, which is in line with the second part of our hypothesis. This difference may partly be mediated by the stronger effect of robusta coffee on the liver expression of two key genes (G6pc and mTOR). As coffee made from robusta species normally contains more caffeine, its larger effect size might be attributed to that phytochemical. Our research paves the way for future mechanistic studies to elucidate the pharmacodynamics of different combinations of bioactive coffee compounds in T2D subjects.

## Figures and Tables

**Figure 1 nutrients-11-02074-f001:**
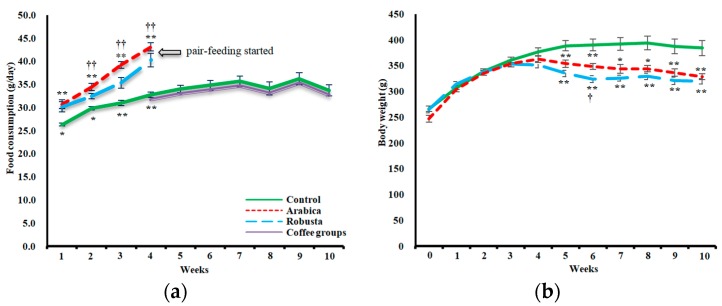
Demonstration of the trend of changes in the average weekly food consumption (**a**) and body weight (**b**) of the three study groups. After starting the pair feeding (week 4), food consumption of arabica and robusta (coffee) groups was matched with the control group. Error bars are SEM. Asterisks denote significantly different values from the control group (* *P* ˂ 0.05, ** *P* ˂ 0.01) and daggers denote significantly different values from the other intervention group († *P* ˂ 0.05, †† *P* ˂ 0.01).

**Figure 2 nutrients-11-02074-f002:**
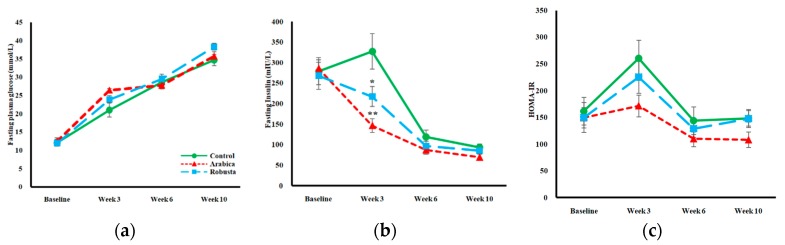
Mean fasting glucose (**a**) and fasting insulin (**b**) of control, arabica, and robusta groups measured every 3rd week. Panel(**c**) shows mean HOMA-IR values calculated from fasting glucose and insulin for each group (glucose mmol/L × insulin mIU/L/22.5). Error bars are SEM. Asterisks denote significantly different values from the control group (* *P* ˂ 0.05, ** *P* ˂ 0.01).

**Figure 3 nutrients-11-02074-f003:**
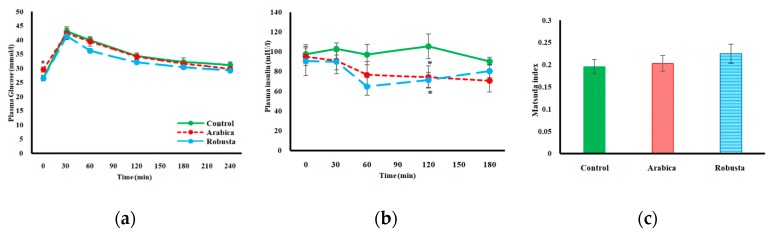
Plasma glucose (**a**) and insulin (**b**) values during an oral glucose tolerance test. Panel (**c**) displays mean Matsuda index of each study group derived from the baseline and average postprandial glucose and insulin values. Error bars are ± SEM. Asterisks denote significantly different values from the control group (* *P* ˂ 0.05).

**Figure 4 nutrients-11-02074-f004:**
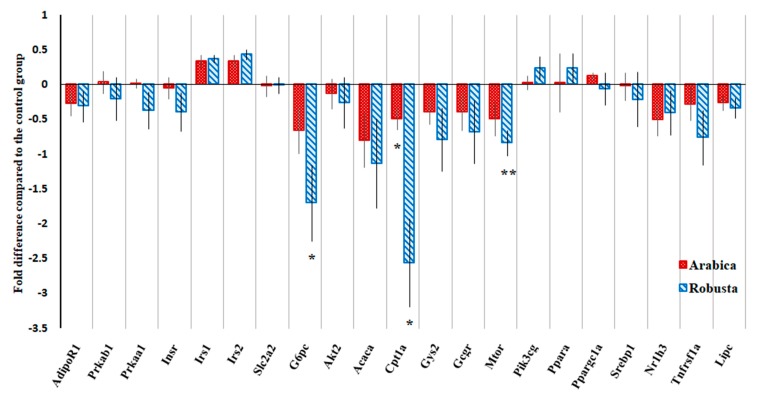
Fold difference in the hepatic expression of selected genes in coffee groups compared to their mean expression in the control group (baseline) calculated by the formula: (ΔCt of coffee groups/mean ΔCt of the control group) − 1. Negative values represent downregulation.

**Table 1 nutrients-11-02074-t001:** List of genes selected for quantitative real-time polymerase chain reaction analysis.

Gene Symbol	Gene Full Name	TaqMan Assay ID
AdipoR1	Adiponectin receptor 1	Rn01483784-m1
Prkab1	Protein kinase AMP-activated non-catalytic subunit beta 1	Rn01499630-m1
Prkaa1	Protein kinase AMP-activated catalytic subunit alfa 1	Rn00665045-m1
Insr	Insulin receptor	Rn01637243-m1
Irs1	Insulin receptor substrate 1	Rn02132493-s1
Irs2	Insulin receptor substrate 2	Rn01482270-s1
Slc2a2	Solute carrier family 2 member 2	Rn00563565-m1
G6pc	Glucose-6-phosphatase, catalytic subunit	Rn00565347-m1
Akt2	AKT serine/threonine kinase 2	Rn00690900-m1
Acaca	Acetyl-CoA carboxylase alpha	Rn00573474-m1
Cpt1a	Carnitine palmitoyltransferase1A	Rn00580702-m1
Gys2	Glycogen synthase 2	Rn00565296-m1
Gcgr	Glucagon receptor	Rn00597158-m1
mTOR	Mechanistic target of rapamycin	Rn00571541-m1
Pik3cg	Phosphatidylinositol-4,5-bisphosphate 3-kinase catalytic subunit gamma	Rn01289357-g1
Ppara	Peroxisome proliferator-activated receptor alfa	Rn00566193-m1
Ppargc1a	Peroxisome proliferator-activated receptor gamma coactivator 1 alpha	Rn00580241-m1
Srebf1	Sterol regulatory element binding transcription factor 1	Rn01495769-m1
Nr1h3	Nuclear receptor subfamily 1, group H, member 3	Rn00581185-m1
Tnfrsf1a	Tumor necrosis factor receptor superfamily, member 1A	Rn01492348-m1
Lipc	Lipase C, hepatic type	Rn00561474-m1
Hprt1	Hypoxanthine phosphoribosyltransferase 1	Rn01527840-m1
Gapdh	Glyceraldehyde-3-phosphate dehydrogenase	Rn01775763-g1

**Table 2 nutrients-11-02074-t002:** Baseline, endpoint, and change of glycated hemoglobin and plasma lipoproteins, and endpoint values of free fatty acids, liver triglyceride, and adiponectin.

		Haemoglobin A_1_C (mmol/mol)	Triglyceride (mmol/L)	Total Cholesterol (mmol/L)	HDL Cholesterol (mmol/L)	LDL Cholesterol (mmol/L)	Free Fatty Acids (mmol/L)	Liver TG Content (mmol/L)	Adiponectin (µg/mL)
**Control**	**Baseline**	20.9 (19.62–22.18)	1.76 (1.25–2.27)	2.90 (2.67–3.12)	1.61 (1.41–1.81)	0.34 (0.26–0.41)	˗	˗	˗
	**Endpoint**	61.8 (55.69–67.91)	2.35 (1.82–2.88)	4.40 (3.97–4.84)	2.96 (2.48–3.43)	0.69 (0.51–0.87)	0.55 (0.46–0.64)	0.79 (0.68–0.90)	4.01 (3.71–4.31)
	**Difference**	40.4 (33.05–47.75)	0.51 (0.41–1.43)	1.51 (1.09–1.92)	1.35 (0.84–1.85)	0.36 (0.18–0.53)	˗	˗	˗
**Arabica**	**Baseline**	22.4 (20.92–23.88)	1.93 (1.43–2.44)	3.23 (2.79–3.67)	1.66 (1.39–1.92)	0.35 (0.22–0.48)	˗	˗	˗
	**Endpoint**	66.6 (62.81–70.39) ††	2.72 (2.23–3.20)	4.63 (4.24–5.02) ††	3.45 (2.95–3.94) †	0.61 (0.49–0.73)	0.57 (0.47–0.66)	0.56 (0.47–0.64) **	4.51 (4.00–5.03)
	**Difference**	43.7 (38.73–48.67) †	0.78 (0.14–1.43)	1.40 (0.67–2.13)	1.79 (1.24–2.33) ††	0.26 (0.04–0.48)	˗	˗	˗
**Robusta**	**Baseline**	22.82 (21.07–24.57)	1.72 (1.49–1.95)	3.11 (2.76–3.47)	1.78 (1.59–1.96)	0.31 (0.20–0.43)	˗	˗	˗
	**Endpoint**	57.27 (53.29–61.26) ††	3.06 (2.41–3.71)	5.45 (4.94–5.95) **††	4.19 (3.59–4.79) **†	0.77 (0.65–0.89)	0.58 (0.50–0.65)	0.62 (0.53–0.71) **	5.08 (4.41–5.74) **
	**Difference**	34.27 (30.22–38.33) †	1.34 (0.65–2.04)	2.33 (1.60–3.07)	2.41 (1.80–3.01) ††	0.46 (0.30–0.61)	˗	˗	˗
**F-test *p*-value**	**Baseline**	0.133	0.773	0.343	0.458	0.875	˗	˗	˗
	**Endpoint**	0.013	0.680	0.002	0.004	0.196	0.902	0.002	0.012
	**Difference**	0.033	0.837	0.056	0.017	0.238	˗	˗	˗

** Significantly different from the control group (*P* ˂ 0.01); † Significantly different from the other intervention group († *P* ˂0.05, †† *P* ˂ 0.01).

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
