# Peer review of "Efficacy of Arabica Versus Robusta Coffee in Improving Weight, Insulin Resistance, and Liver Steatosis in a Rat Model of Type-2 Diabetes"

_nutrients, 2019, doi:10.3390/nu11092074_

Round 1
Reviewer 1 Report
This study uses a widely accepted rat model of T2D to test the hypotheses that long-term coffee consumption has benefits for controlling T2D and associated metabolic disturbances and that the efficacy of 2 coffee species will differ. The data presented appear to be well carried out and some marginal changes in lipid profile were observed. However, there are issues with interpretation of the data. First, the increased food consumption in the coffee fed groups, which necessitated pair feeding to be added to the study design, is problematic. This suggests either an increase in basal metabolic rate or increased activity in response to coffee. It is stated in the discussion that rats were “compensating for their higher energy demands by overeating” without explanation as to why their energy demands would be higher. ZDF rats overeat anyway, so this aspect does require clarification. Neither metabolic rate nor activity levels of the rats were measured making further interpretation difficult. The observed changes in lipid and insulin levels may well be attributable to such secondary responses rather than directly due to components of the coffee.
Differences in response to robusta and arabica coffee are of interest, however interpretations of the differences in this study are difficult to make as no chemical analysis of the coffee was performed. Were there differences in caffeine, phenolic or alkaloid levels in the preparations used?
Improvements in insulin sensitivity appear to be minimal or non-existent, with only a small decrease in fasting insulin 2 hours after an oral glucose challenge and no difference in Matsuda index. HOMA is mentioned in the methods (line 98), results (line 181) and discussion (line 273 to 279) and "a tendency to lower HOMA-IR levels" is stated, but the HOMA-IR values are not shown in the figures or tables or given in the results. There appears to have been something overlooked here.
The differences in gene expression, while interesting, are difficult to interpret given the issues raised above.
Author Response
To
Reviewer 1
Thank you for your comments. Your comments have been discussed by the authors and HOMA-IR results are now displayed in figure 2c.
Comment: This study uses a widely accepted rat model of T2D to test the hypotheses that long-term coffee consumption has benefits for controlling T2D and associated metabolic disturbances and that the efficacy of 2 coffee species will differ. The data presented appear to be well carried out and some marginal changes in lipid profile were observed. However, there are issues with interpretation of the data. First, the increased food consumption in the coffee fed groups, which necessitated pair feeding to be added to the study design, is problematic. This suggests either an increase in basal metabolic rate or increased activity in response to coffee. It is stated in the discussion that rats were “compensating for their higher energy demands by overeating” without explanation as to why their energy demands would be higher. ZDF rats overeat anyway, so this aspect does require clarification. Neither metabolic rate nor activity levels of the rats were measured making further interpretation difficult.
Answer: Regarding pair-feeding, this procedure was chosen as a scientific compromise in order to avoid to large differences in food intake between the groups which per se could negatively impact the diabetic conditions with regards to body weight increase. Pair feeding is a widely used method in animal studies[1]. Besides, to our knowledge, no studies have examined the effect of coffee in ZDF rats. We have now added further considerations in the discussion section (lines 266-270).
Comment: The observed changes in lipid and insulin levels may well be attributable to such secondary responses rather than directly due to components of the coffee.
Answer: Thank you for this relevant input. We acknowledge that these differences could be caused by the lower body weight. However, the fact that the animals in the pair feeding period have equal food intake indicate that the differences are in fact direct effects of coffee.
Comment: Differences in response to robusta and arabica coffee are of interest, however interpretations of the differences in this study are difficult to make as no chemical analysis of the coffee was performed. Were there differences in caffeine, phenolic or alkaloid levels in the preparations used?
Answer: We do not have any analyses regarding chemical composition of the coffee species used in this study. However, it is already known that the two species contain different amounts of trigonelline, caffeine and chlorogenic acid as stated in line 41
Comment: Improvements in insulin sensitivity appear to be minimal or non-existent, with only a small decrease in fasting insulin 2 hours after an oral glucose challenge and no difference in Matsuda index. HOMA is mentioned in the methods (line 98), results (line 181) and discussion (line 273 to 279) and "a tendency to lower HOMA-IR levels" is stated, but the HOMA-IR values are not shown in the figures or tables or given in the results. There appears to have been something overlooked here.
Answer: You are right, thank you for your observation. HOMA-IR results are now illustrated (figure 2c, line 190).
Comment: The differences in gene expression, while interesting, are difficult to interpret given the issues raised above.
Answer: With reference to our answer above the restriction on food intake was a scientific compromise and this may of course impact on the gene expression profile. However, we still think that it was the right decision to induce and maintain equal food between groups.
On behalf of the authors,
Sincerely
Christine Bodelund Christiansen
Research assistant, Aarhus University
Palle Juul-Jensens boulevard 165
8200 Aarhus
Denmark
E-mail:CHBOCR@rm.dk, Mobile:+45 28518032
[1] References:
1) Cade, C.; Norman, AW. Vitamin D3 improves impaired glucose tolerance and insulin secretion in the vitamin D-deficient rat in vivo.Endocrinology, 1986, 119, 84-90, doi: 10.1210/endo-119-1-84.
2) Brandt, A.; Berghein, I et al. Consumption of decaffeinated coffee protects against the development of early non-alcoholic steatohepatitis: Role of intestinal barrier function. Redoxbiol., 2019, 21, 101092, doi: 10.1016/j.redox.2018.101092.
3) Furuhata, Y.; Nishihara, N. et al. Effects of pair-feeding and growth hormone treatment on obese transgenic rats. Eur J Endocrinol., 2002, 146, 244-49, doi: 10.1530/eje.0.1460245.

Reviewer 2 Report
Title: Efficacy of Arabica versus Robusta Coffee in Improving Weight, Insulin Resistance, and Liver Steatosis in a Rat Model of Type 2 Diabetes
The authors presented a study that was quite interesting and I do believe this particular topic could also spark the interest of those reading the journal Nutrients. However, I have several concerns within the manuscript that limit the interpretability of the findings and ultimately the contribution that this paper can make to the literature in its current form. The authors will need to consider the following comments:
Introduction, paragraph 3. The authors do NOT provide a sufficient review of the literature to better frame the study hypotheses and their eventual findings. Later in the discussion section the authors reference studies [14-18], some of which appear to be related to the current investigation. For example, reference 14 is also a study on "Influence of coffee brew in metabolic syndrome and T2D". As a reader, I would like to know more about this investigation [i.e., reference 14], since it is similar to the current investigation. Subsequently, the authors need to provide a sufficient review of these references in the introduction section of the paper, which will enhance its overall readability.
Method, Study Design and Procedures. The authors are asked to simply elucidate why instant coffee was utilized in this investigation as opposed to other forms of coffee (e.g., coarse coffee grounds or condensed tablets). While this may be some form of common practice or preferred method of coffee distribution in animal/rodent models, this was just NOT apparent to this reviewer, and thus, future readers of this particular study may also share in this question. Please add this clarification to your methods section.
Table 2. In the current manuscript, Table 2 appears to be missing some form of table key to indicate to readers various significance levels. This will need to be added to Table 2 and also to Table 4 (here the "*" and "**" do not have a description of significance). Although most readers would be able to interpret the significant levels on their own, I believe the authors do not want this to look sloppy or that minor details such as this go overlooked. In addition, the pagination of the lines within Table 2 are misaligned, which made interpreting the results of this table quite difficult and also reflects my point noted above.
Discussion Section. In the third paragraph of the discussion section, the information regarding the short-term studies of the KK-Ay mice [17] and Zucker fatty rats [14] needs to be expanded as this appears to be an important point the authors are making.
Minor Revisions
Method, Study Design and Procedures. In the second paragraph, the sentence beginning at line [88], "Rats fasted for 6 to 8 h before each blood sampling procedure including the OGTT," appeared to be out of place and could be moved to an earlier section of the paragraph, to enhance the overall readability of this point. I do not disagree with the authors' claims, just that it should be repositioned.
Results, Glucose Homeostasis. Please make an addition at the end of line 174 indicating that "rats receiving robusta coffee, however, had a lower increase in HbA1c levels throughout the study in comparison with Arabica group (Table 2), THOUGH AGAIN THESE FINDINGS ARE NON-SIGNIFICANT".

Author Response
To
Reviewer 2
Thank you for your comments. Your comments have been discussed by the authors. Accordingly, reference 14-18 have been elaborated in more detail, the reason for using instant coffee has been described and a table key have been added to table 2.
Title: Efficacy of Arabica versus Robusta Coffee in Improving Weight, Insulin Resistance, and Liver Steatosis in a Rat Model of Type 2 Diabetes
The authors presented a study that was quite interesting and I do believe this particular topic could also spark the interest of those reading the journal Nutrients. However, I have several concerns within the manuscript that limit the interpretability of the findings and ultimately the contribution that this paper can make to the literature in its current form. The authors will need to consider the following comments:
Comment: Introduction, paragraph 3.The authors do NOT provide a sufficient review of the literature to better frame the study hypotheses and their eventual findings. Later in the discussion section the authors reference studies [14-18], some of which appear to be related to the current investigation. For example, reference 14 is also a study on "Influence of coffee brew in metabolic syndrome and T2D". As a reader, I would like to know more about this investigation [i.e., reference 14], since it is similar to the current investigation. Subsequently, the authors need to provide a sufficient review of these references in the introduction section of the paper, which will enhance its overall readability.
Answer: Thank you for your relevant point, we agree. The studies [14-18] have been described in more detail regarding study design and findings (lines 48-60):
" Several short term studies have indicated beneficial effects of coffee or coffee components in regards to T2D. Thirty days of administration of coffea arabica to Zucker fa/fa improved triglycerides, low-density lipoprotein (LDL) cholesterol, and glucose levels compared to Zucker fa/fa control rats, with the levels corresponding to the levels in Zucker +/+ rats. However, glucose levels improved only at intervention week 3 and a reduction in low-density lipoprotein (HDL) cholesterol was found [6]. Also, 5 weeks intervention with coffee administered to KK-ay mice lowered blood glucose and liver lipids compared to water [7]. Furthermore, lepr db/db mice treated for 2 weeks with chlorogenic acid showed glucose, insulin, and lipid levels similar to healthy rats and metformin-treated lepr db/db mice. Chlorogenic acid also improved glucose uptake in skeletal muscle and lower liver lipid content was found[8]. Similarly, chlorogenic acid improved levels of blood and liver lipids in comparison to control solution in a 3-weeks intervention study using Zucker fa/fa rats [9]. Also caffeine [7] as well as trigonelline and nicotinic acid [10] have beneficial effects toward T2D shown by lower blood glucose and liver lipids levels in KK-ay mice."
Comment: Method, Study Design and Procedures. The authors are asked to simply elucidate why instant coffee was utilized in this investigation as opposed to other forms of coffee (e.g., coarse coffee grounds or condensed tablets). While this may be some form of common practice or preferred method of coffee distribution in animal/rodent models, this was just NOT apparent to this reviewer, and thus, future readers of this particular study may also share in this question. Please add this clarification to your methods section.
Answer: Thank you, relevant point. We used instant coffee as an alternative to filter coffee as the instant coffee is easier to handle since it was possible to incorporate it into the food rather than adding it to the drinking water. It should be noticed that filter coffee is the most widely consumed form of coffee in Scandinavia. It has now been added at lines 87-89 in the method section.
Comment: Table 2. In the current manuscript, Table 2 appears to be missing some form of table key to indicate to readers various significance levels. This will need to be added to Table 2 and also to Table 4 (here the "*" and "**" do not have a description of significance). Although most readers would be able to interpret the significant levels on their own, I believe the authors do not want this to look sloppy or that minor details such as this go overlooked. In addition, the pagination of the lines within Table 2 are misaligned, which made interpreting the results of this table quite difficult and also reflects my point noted above.
Answer: Thank you for your observation. The various significant levels are now described in line 196. We have added gridlines in the figure for easier interpretation.
Comment: Discussion Section. In the third paragraph of the discussion section, the information regarding the short-term studies of the KK-Ay mice [17] and Zucker fatty rats [14] needs to be expanded as this appears to be an important point the authors are making.
Answer: Following your suggestions, this information has now been added in the introduction.
Minor Revisions
Comment: Method, Study Design and Procedures. In the second paragraph, the sentence beginning at line [88], "Rats fasted for 6 to 8 h before each blood sampling procedure including the OGTT," appeared to be out of place and could be moved to an earlier section of the paragraph, to enhance the overall readability of this point. I do not disagree with the authors' claims, just that it should be repositioned.
Answer: Agree, the sentence have been removed from line 105 to line 100 (in the revised version).
Comment: Results, Glucose Homeostasis. Please make an addition at the end of line 174 indicating that "rats receiving robusta coffee, however, had a lower increase in HbA1c levels throughout the study in comparison with Arabica group (Table 2), THOUGH AGAIN THESE FINDINGS ARE NON-SIGNIFICANT".
Answer: There is a significant lower increase in HbA1c in the robusta group in comparison with the arabica group as indicated by a significant difference (p=0.033) which is marked by "†" in the figure. The legends for table has been revised to clarify this.
On behalf of the authors,
Sincerely
Christine Bodelund Christiansen
Research assistant, Aarhus University
Palle Juul-Jensens boulevard 165
8200 Aarhus
Denmark
E-mail:CHBOCR@rm.dk, Mobile:+45 28518032

Round 2
Reviewer 1 Report
The authors are to be congratulated for making improvements to the manuscript and have now included the HOMA data. However, I still have a major concern with the lack of chemical analysis of the coffee. The reference cited to support the differences between robusta and arabica coffee (ref 3. Martin et al. 1998) does show ON AVERAGE higher caffeine and chlorogenic acid levels in robusta, but also identifies substantial variation such that caffeine and chlorogenic acid content of 2 of the 13 robusta varieties tested were within the range expected for arabica. Further, none of the robusta varieties tested in reference 3 originated from Brazil. Thus it cannot be assumed that chlorogenic acid and caffeine contents of the robusta coffee used in this study will be higher than the arabica. As a consequence the conclusion (lines 371 to 373) "As coffee made from robusta species normally contains more caffeine and chlorogenic acids, its larger effect size can be attributed to either or both of these phytochemicals" can not be substantiated unless caffeine and chlorogenic acid levels are measured. It is surprising to me that this information (at least for the caffeine) is not available given that the coffee was sourced from a major manufacturer of these products.
I would also like to make a further suggestion for the discussion (lines 284 to 289). One interpretation of the lack of effect on FPG beyond 3 weeks is a failure of insulin production (figure 2b) which is a known characteristic of ZDF rats that does not occur in Zucker fatty rats (Tokuyama et al. Diabetes 44:1447, 1995). At week 3 the elevated fasting glucose and insulin values are consistent with insulin resistance with compensatory increased insulin secretion. At week 6 FPG is further elevated yet insulin has declined, suggestive of beta cell failure. The authors should consider this as a possible alternative explanation.
Author Response
To
Reviewer 1
Comment: The authors are to be congratulated for making improvements to the manuscript and have now included the HOMA data. However, I still have a major concern with the lack of chemical analysis of the coffee. The reference cited to support the differences between robusta and arabica coffee (ref 3. Martin et al. 1998) does show ON AVERAGE higher caffeine and chlorogenic acid levels in robusta, but also identifies substantial variation such that caffeine and chlorogenic acid content of 2 of the 13 robusta varieties tested were within the range expected for arabica. Further, none of the robusta varieties tested in reference 3 originated from Brazil. Thus it cannot be assumed that chlorogenic acid and caffeine contents of the robusta coffee used in this study will be higher than the arabica.
Answer: Thank you for your relevant comment. The introduction has now been updated to make it clear that chemical composition in each coffee species can vary. We have also added additional references on the chemical composition of Brazilian coffee (lines 42-43).
Comment: As a consequence the conclusion (lines 371 to 373) "As coffee made from robusta species normally contains more caffeine and chlorogenic acids, its larger effect size can be attributed to either or both of these phytochemicals" can not be substantiated unless caffeine and chlorogenic acid levels are measured. It is surprising to me that this information (at least for the caffeine) is not available given that the coffee was sourced from a major manufacturer of these products.
Answer: We have been in contact with the manufacturer and they have provided the caffeine content of the instant coffee as 2.35 and 3.85% of dry weight for arabica and robusta, respectively, which has been added to the method section (lines 91-94). We have deleted chlorogenic acid from the line in the discussion (lines 382-383).
Comment: I would also like to make a further suggestion for the discussion (lines 284 to 289). One interpretation of the lack of effect on FPG beyond 3 weeks is a failure of insulin production (figure 2b) which is a known characteristic of ZDF rats that does not occur in Zucker fatty rats (Tokuyama et al. Diabetes 44:1447, 1995). At week 3 the elevated fasting glucose and insulin values are consistent with insulin resistance with compensatory increased insulin secretion. At week 6 FPG is further elevated yet insulin has declined, suggestive of beta cell failure. The authors should consider this as a possible alternative explanation.
Answer: Thank you for this very relevant observation. We have now added this consideration to our discussion section (lines 289-296).
On behalf of the authors,
Sincerely
Christine Bodelund Christiansen
Research assistant, Aarhus University
Palle Juul-Jensens boulevard 165
8200 Aarhus
Denmark
E-mail:CHBOCR@rm.dk, Mobile:+45 28518032
